# Antimicrobial Activity of Six International Artisanal Kefirs against *Bacillus cereus*, *Listeria monocytogenes*, *Salmonella enterica* Serovar Enteritidis, and *Staphylococcus aureus*

**DOI:** 10.3390/microorganisms8060849

**Published:** 2020-06-04

**Authors:** Abrar Sindi, Md. Bahadur Badsha, Barbara Nielsen, Gülhan Ünlü

**Affiliations:** 1School of Food Science, University of Idaho, 875 Perimeter Drive, MS 2312, Moscow, ID 83844-2312, USA; sind3904@vandals.uidaho.edu (A.S.); barbn@uidaho.edu (B.N.); 2Institute for Modeling Collaboration and Innovation (IMCI), University of Idaho, 875 Perimeter Drive, MS 1122, Moscow, ID 83844-1122, USA; mdbadsha@uidaho.edu; 3School of Food Science, Washington State University, Pullman, WA 99164-6376, USA; 4Department of Biological Engineering, University of Idaho, 875 Perimeter Drive, MS 2312, Moscow, ID 83844-0904, USA

**Keywords:** artisanal kefir, kefir product, kefir grain, natural antimicrobial, bacteriocin, *Listeria monocytogenes*, *Salmonella enterica* serovar Enteritidis, *Staphylococcus aureus*, *Bacillus cereus*

## Abstract

Kefir, a fermented dairy beverage, exhibits antimicrobial activity due to many metabolic products, including bacteriocins, generated by lactic acid bacteria. In this study, the antimicrobial activities of artisanal kefir products from Fusion Tea (A), Britain (B), Ireland (I), Lithuania (L), the Caucuses region (C), and South Korea (K) were investigated against select foodborne pathogens. *Listeria monocytogenes* CWD 1198, *Salmonella enterica* serovar Enteritidis ATCC 13076, *Staphylococcus aureus* ATCC 25923, and *Bacillus cereus* ATCC 14579 were inhibited by artisanal kefirs made with kefir grains from diverse origins. Kefirs A, B, and I inhibited all bacterial indicator strains examined at varying levels, except *Escherichia coli* ATCC 12435 (non-pathogenic, negative control). Kefirs K, L, and C inhibited all indicator strains, except *S. aureus* ATCC 25923 and *E. coli* ATCC 12435. Bacteriocins present in artisanal kefirs were determined to be the main antimicrobials in all kefirs examined. Kefir-based antimicrobials are being proposed as promising natural biopreservatives as per the results of the study.

## 1. Introduction

Foodborne illnesses represent a significant public health challenge worldwide, with almost 1 in 10 people becoming sick and 33 million people dying [1]. Foodborne pathogens also have a huge impact on the economy. According to the World Bank Organization, the total productivity loss related to foodborne disease in low- and middle-income countries is valued at $95.2 billion per year, in addition to the annual cost of $15 billion used to treat affected individuals [2].

The Center for Disease Control and Prevention (CDC) estimates numbers for foodborne illness each year in the United States at 47.8 million cases, with 128,000 hospitalizations, 3030 deaths, and $78 billion in cost, including the costs attributed to premature deaths, medical expenses, and loss of productivity. Unspecified pathogens cause 80% of the illnesses (38.4 million illnesses, 72,000 hospitalizations, and 1686 deaths) [3]. Known pathogens are responsible for 20% of the illnesses, estimated at 9.4 million illnesses, 59,961 hospitalizations, and 1351 deaths [4]. Of these, 90% have been linked to just seven microorganisms: *Campylobacter* sp., *Clostridium perfringens*, *Escherichia coli* (Serotype O157:H7)*, Listeria monocytogenes, Salmonella* (non-typhoidal), norovirus, and *Toxoplasma gondii*. Non-typhoidal *Salmonella* was determined to be the principal cause of hospitalization and death among these seven pathogens [5].

Disability-adjusted life year (DALY) is a measure developed by the World Health Organization [1]. The DALY pools data on premature mortality and morbidity from acute illness and long-term sequelae into a single statistic, which in turn synopsizes years of healthy life lost. Scallan et al. [5] explored the overall impact of foodborne illness caused by the seven leading foodborne pathogens in the United States using DALY. They defined health states (acute illness and long-term sequelae) for each foodborne pathogen and then estimated the average annual incidence of each health state using data from public health surveillance and previously published estimates. These seven foodborne pathogens caused about 112,000 DALYs on an annual basis due to foodborne illnesses acquired in the United States. Non-typhoidal *Salmonella* (32,900) and *Toxoplasma gondii* (32,700) caused the most DALYs, trailed by *Campylobacter* sp. (22,500), norovirus (9900), *L. monocytogenes* (8800), *C. perfringens* (4000), and *E. coli* O157:H7 (1200). Among all foodborne pathogenic bacteria that can cause foodborne illness, non-typhoidal *Salmonella* and *L. monocytogenes* are responsible for 42,900 DALYs total (37% of all DALYs). These two organisms are included as target organisms in the presented work. As a foodborne pathogen, *Bacillus cereus* is estimated to cause 63,623 foodborne disease cases per year in the United States [4,6]. Staphylococcal food poisoning accounts for about 241,994 foodborne disease cases per year in the United States [6]. *B. cereus* and *S. aureus* are also included in our work because foodborne illness caused by these organisms are highly underreported and underdiagnosed [6].

Reducing foodborne illness takes a great deal of time, effort, and collaboration. The ultimate goal for most public health and food safety officials worldwide is not just stopping foodborne illness outbreaks once they occur but also preventing them from happening in the first place. Long-term prevention of foodborne illness outbreaks takes the actions of countless partners in the food production chain stretching from farm to table: production, harvest, storage, processing, distribution, and preparation. Chemical preservation, biologically based preservation, and physical methods of food preservation are all used, individually and in combination, to inhibit foodborne pathogens in food processing.

Biologically based preservation methods are among the newer and emerging forms of food preservation. According to Matthews et al., “biopreservation is the use of lactic acid bacteria (LAB), their metabolic products, or both to improve or ensure the safety and quality of products that are not fermented” [7,8]. Some LAB produce antimicrobial peptides called bacteriocins which have been shown to inhibit foodborne pathogenic bacteria. Bacteriocins are ribosomally synthesized proteins or peptides that are secreted by bacteria that inhibit other closely related bacteria using various mechanisms [8,9]. Bacteriocins are divided into four major groups: class I, class II, class III, and class IV. The class I bacteriocins (lantibiotics) contain the first bacteriocin discovered from LAB, nisin [10]. Bacteriocins such as nisin are safe for human consumption since they are natural proteins and peptides that are degraded by the digestive enzymes in the stomach [11]. Nisin has “generally recognized as safe” (GRAS) status in the United States, granted by the United States Food and Drug Administration (FDA), for several applications in the food industry. It has been added as a food safety measure to a variety of foods in the world market, including dairy products, canned foods, salad dressings, sauces, and baby food. Nisin is effective against Gram-positive pathogens such as *S. aureus*, *B. cereus*, *L. monocytogenes,* and *C. perfringens* [12,13].

Kefir, a fermented dairy beverage produced by the actions of the microflora encased in the “kefir grain” on the carbohydrates in the milk, originated thousands of years ago in the Caucasus mountain region between Europe and Russia. Containing many bacterial species already known for their probiotic properties, it has long been popular in Eastern Europe for its purported health benefits, where it is routinely administered to patients in hospitals and recommended for infants and the infirm. More than 30–50 species of yeasts (*Saccharomyces* sp., *Kluyveromyces* sp., *Candida* sp., *Torulaspora* sp., *Cryptococcus* sp., *Pichia* sp., etc.) and LAB (*Lactobacillus* sp., *Lactococcus* sp., *Leuconostoc* sp., etc.) have been isolated and identified from kefir grains [14,15]. Kefir grains have been shown to have regional differences in microbial composition, producing variability in the kefirs produced, due in part to local LAB finding a niche in the grains [16].

Kefir has been shown to contain a variety of natural antimicrobials, including bacteriocins, organic acids, hydrogen peroxide, and fatty acids. A Brazilian kefir product showed growth inhibition, measured as percent inhibition, against *S. aureus* ATCC 6538 (42.80%–69.15%), *E. coli* ATCC 11229 (30.73%–59.89%), *S. typhi* ATCC 6539 (44.99%–73.05%), *L. monocytogenes* ATCC 15313 (41.45%–54.18%) and *B. cereus* RIBO 1222-173-S4 (70.38%–86.80%) [17]. Another study investigated the antimicrobial activity of a Romanian kefir product against *B. subtilis* spp. *spizizenii* ATCC 6633, *S. aureus* ATCC 6538, *E. coli* ATCC 8739 *, Enterococcus faecalis* ATCC 29212 and *S. enteritidis* ATCC 13076. The Romanian kefir showed strong antibacterial activity against Gram-negative and Gram-positive indicator strains when compared to neomycin sulfate and ampicillin [18]. The antimicrobial spectra of four types of kefirs (A, L, M, and S) from South Korea were determined in another study. With kefir A, *B. cereus* ATCC 14579, *E. coli* ATCC 25922, *S. enterica* serovar Enteritidis FDA, *Pseudomonas aeruginosa* ATCC 15522, and *Cronobacter sakazakii* ATCC 29544 were inhibited. *B. cereus* ATCC 14579, *S. aureus* ATCC 6538, *E. coli* ATCC 25922, *S. enterica* serovar Enteritidis FDA, *P. aeruginosa* ATCC 15522, and *C. sakazakii* ATCC 29544 were inhibited to different extents by kefirs L, M, and S. *L. monocytogenes* ATCC 51776 was only inhibited by kefir M [19].

To our knowledge, comparisons among international artisanal kefirs regarding their antimicrobial activities against select foodborne pathogens have not been reported. For this study, we hypothesized that international artisanal kefirs have diverse microflora, generating distinctive bacteriocin content, resulting in varied levels of antimicrobial activities. The objectives of our study were to 1) compare the antimicrobial activity of artisanal kefirs from Fusion Tea (A), Britain (B), the Caucuses region (C), Ireland (I), Lithuania (L), and South Korea (K) against select foodborne pathogens, and 2) examine whether the antimicrobial effect is due to bacteriocin production or other antimicrobials present in kefir. 

## 2. Materials and Methods

### 2.1. Artisanal Kefir Preparation for Determining Kefir Antimicrobial Activity

Six types of artisanal kefir grains originating from Britain (B; Etsy Inc., Brooklyn, New York, NY, USA), the Caucuses (C; Etsy Inc., Brooklyn, New York, NY, USA), Ireland (I; Etsy Inc., Brooklyn, New York, NY, USA), Lithuania (L; Etsy Inc., Brooklyn, New York, NY, USA), South Korea (K) [19], and a compilation of blended world-sourced grains (A; Fusion Tea, Amazon, Seattle, WA, USA) were used in this study. Kefir grains were examined to evaluate their similarities and differences in shape, appearance, texture, and size while kefir products were evaluated for their flavor, aroma, and texture. 

Artisanal kefir products were prepared using traditional methods. Kefir grains were inoculated into pasteurized whole milk daily for at least one week before any experiment. Kefir grains were inoculated (10% (w/v)) into whole pasteurized milk and the resulting mixture was incubated at 22–24 °C for 24 h. The fermentation process was stopped when the pH reached 3.9–4.1 as measured by a calibrated PB Basic Meter (Denver Instrument, Bohemia, NY, USA). A clean plastic strainer with 1 mm pore size was used to separate the grains from kefir products. Designated plastic strainers were used for each kefir product/grains to avoid cross-contamination. Once separated, the kefir grains were inoculated (10% (w/v)) into a new batch of whole pasteurized milk to maintain their activity. The resulting kefir products were centrifuged at 10,000× *g* for 15 min at 4 °C using an Avanti J-17 high-speed centrifuge (Beckman Coulter, Inc., Palo Alto, CA, USA) to remove solids. Filtration through a Millex filter (Millipore Corporation, Bedford, MA, USA) with 45 µm pore size was used to sterilize the samples. The filter-sterilized kefir samples were tested for their antimicrobial activities immediately.

### 2.2. Protein Concentration Measurements

The total protein concentration was used as a measure to standardize artisanal kefir samples. A Bio-Rad Protein Assay Kit (Bio-Rad Laboratories, Hercules, CA, USA) was used to determine the total protein concentration in kefir samples. A standard procedure for microtiter plates was used with bovine serum albumin (standard II). Absorbance was measured at 595 nm using a microtiter plate reader (SpectraMax 190, Molecular Devices, San Jose, CA, USA). Kefir samples were examined in triplicate to determine the total protein concentration.

### 2.3. Bacterial Strains, Microbiological Media, and Growth Conditions

Bacterial strains used in the study were obtained from American Type Culture Collection (ATCC; Manassas, VA, USA), National Collection of Dairy Organisms (NCDO; now National Collection of Food Bacteria (NCFB); Scotland) and our in-house culture collection. *Lactobacillus plantarum* NCDO 995 and *Micrococcus luteus* ATCC 10420 were used as the non-pathogenic indicator strains. *M. luteus* was selected for its sensitivity to bacteriocins [20]. *Lb. plantarum* is also sensitive to bacteriocins produced by closely related LAB [21]. *E. coli* ATCC 12435 is not sensitive to bacteriocins from LAB and thus was used as a negative control. The pathogenic indicator strains included *S. enterica* serovar Enteritis ATCC 13076, *S. aureus* ATCC 25923, *B. cereus* ATCC 14579, and *L. monocytogenes* CWD 1198. Frozen stocks were maintained in sterile glycerol (25%) and tryptic soy broth (TSB, Criterion, Hardy Diagnostics, Santa Maria, CA, USA) or Lactobacillus MRS broth (Remel, Thermo Fisher Scientific, Lenexa, KS, USA) and kept at –80 °C. Prior to experiments, all indicator organisms were streaked onto tryptic soy agar (TSA) or Lactobacillus MRS agar (for *Lb. plantarum* NCDO 995) and incubated at their optimum growth temperature. The following selective media were used to streak the pathogenic indicators for isolation as needed: Xylose Lysine Deoxycholate agar (XLD) for *S. enterica* serovar Enteritis ATCC 13076, Mannitol Yolk Polymyxin (MYP) agar for *B. cereus* ATCC 14579, Baird Parker agar (BPA) for *S. aureus* ATCC 25923, and HardyCHROM Listeria for *L. monocytogenes*. XLD, MYP, BPA, and HardyCHROM Listeria were obtained from Hardy Diagnostics (Santa Maria, CA, USA). The resulting single colonies on selective media were picked and inoculated into TSB and incubated for 24 h. The incubation temperature used for *Lb. plantarum* NCDO 995, *S. enterica* serovar Enteritis ATCC 13076, *S. aureus* ATCC 25923, and *L. monocytogenes* CWD 1198 was 35–37 °C. *B. cereus* ATCC 14579 and *M. luteus* ATCC 10420 cultures were incubated at 30 °C. *M. luteus* ATCC 10420 was incubated while shaking at 200 rpm in an orbital shaking incubator.

### 2.4. Detection of Antimicrobial Activity in Artisanal Kefirs

The agar-well-diffusion method [20] with some modifications was used to study the antimicrobial activity of filter-sterilized artisanal kefir samples. Soft Lactobacillus MRS and TSA, containing 0.75% agar, were inoculated (10^7^–10^8^ CFU/mL) with the indicator (non-pathogenic and pathogenic) organisms. Bacterial growth curves, generated for all indicator organisms, were used for determining accurate inoculum levels. Filter-sterilized kefir samples were obtained as described above. The wells generated with sterile plastic Pasteur pipettes in soft agar were filled with 100, 150, 200, and 250 µL of filter-sterilized kefir samples. Nisin (1000 IU/mg) and polylysine (1200 IU/mg), purchased from Zhengzhou Bainafo Bioengineering Co., Ltd. (Zhengzhou City, China), were used as positive bacteriocin controls at 220 IU and 200 IU, respectively, and in a volume of 100 µL. Sterilized distilled water (DI) was used as a negative control, also in a volume of 100 µL. Lactobacillus MRS and TSA plates with wells containing filter-sterilized kefir samples, bacteriocin controls, and sterile DI water were incubated for 24 h at 30 °C or 37 °C depending on the optimum growth temperature of the indicator strains. Following the incubation, the diameter of clear zones was measured (in mm) using a ruler. Two independent experiments, each in duplicate, were performed for any given kefir sample.

### 2.5. Ruling-Out Any Antimicrobial Activity Due to Organic Acids, Hydrogen Peroxide, and Free Fatty Acids Produced in Artisanal Kefir

#### 2.5.1. Artisanal Kefir Preparation

The Fusion Tea, Irish, and South Korean kefirs (A, I, and K) were selected for this study due to their high antibacterial activity observed in experiments using the agar-well-diffusion method. Artisanal kefir samples were prepared as mentioned above (2.1).

#### 2.5.2. Bacterial Strains, Microbiological Media, and Growth Conditions

Bacterial strains, microbiological media, and growth conditions were the same as described above.

#### 2.5.3. Detection of Antimicrobial Activity Due to Bacteriocin Production in Artisanal Kefir

The agar-well-diffusion method was used to detect antimicrobial activity due to bacteriocins present in artisanal kefir samples following the protocol by Dimitrieva-Moats and Ünlü (2011) and Ünlü et al. (2015) [20,21] with some modifications. Artisanal kefir samples were prepared as mentioned above, filter sterilized, and pH adjusted to 6.0 using NaOH (5M). Sterile bovine liver catalase (2000–5000 U/mg protein), *Aspergillus oryzae* lipase (≥ 100,000 U/g), and Proteinase K from *Tritirachium album* (≥ 30 units/mg), all of which were purchased from Sigma Aldrich (Saint Louis, MO, USA), were added to sterile kefir samples A, K, and I at a final concentration of 1 mg/mL. β-glycerophosphate (Sigma Aldrich, Saint Louis, MO, USA) was added to sterile kefir samples A, K, and I at 1% (w/v). These mixtures were incubated at 37 °C for 1 h. Sterile β-glycerophosphate was used to buffer artisanal kefir samples with pH adjusted to 6.0. Bovine liver catalase and lipase from *Aspergillus oryzae* were used to degrade hydrogen peroxide and free fatty acids, respectively. Proteinase K from *Tritirachium album* (≥ 30 units/mg) was used to break down bacteriocins, which are proteinaceous, and thus confirm their contribution to antimicrobial activity in kefir. Indicator strains (100 µL of cultures containing 10^8^–10^9^ CFU/mL) were spread onto soft TSA agar (0.75 % agar (w/v)). Several wells with a diameter of 7 mm were formed with sterile plastic Pasteur pipettes and the filter-sterilized kefir samples (100 µL), with and without treatments, were added to the wells. Nisin and polylysine were used as positive bacteriocin controls at 220 IU and 200 IU, respectively, and in a volume of 100 µL. Sterilized distilled water (DI) was used as a negative control in the volume of 100 µL. Plates were incubated overnight at 30 °C or 37 °C depending on the indicator strain. The diameter (in mm) of clear zones was measured using a ruler. Two independent experiments, each in duplicate, were performed for any given kefir sample.

### 2.6. Statistical Analysis

For the detection of the antimicrobial activity in artisanal kefirs, the experiment was a completely random design. A three-way ANOVA was used for the following: indicator organisms, kefir types, kefir volumes and their interactions, followed by Tukey’s multiple comparison procedure using R (R Studio Inc., Boston, MA, USA). For ruling out any antimicrobial activity due to organic acids, hydrogen peroxide and free fatty acids produced in kefir, a three-way ANOVA was used as well, followed by Tukey’s multiple comparison procedure using R. The statistically significant difference was determined by cut-off for significance level at 5% (i.e., *p* < 0.05).

## 3. Results

### 3.1. Artisanal Kefir Products and Kefir Grains Description

Artisanal kefir products and grains were described in Table 1. The kefir grains sizes ranged from >1 mm to 50 mm depending on the origin. The smallest grains were from Ireland while the largest grains were from Britain. Kefir products differed in their flavor and aroma, as indicated in Table 1. 

### 3.2. The Antimicrobial Activity Spectra of Filter-Sterilized Artisanal Kefirs

#### 3.2.1. Protein Concentration

All artisanal kefir samples tested were standardized based on total protein content (0.5 mg/mL kefir).

#### 3.2.2. The Antimicrobial Activity Spectra of Filter-Sterilized Artisanal Kefirs Determined by the Agar Well Diffusion Method 

As anticipated, nisin and polylysine controls inhibited the growth of all pathogenic and non-pathogenic indicator organisms excluding *E. coli* ATCC 12435, a non-pathogenic indicator organism used as the negative control (Figure 1a–g). Nisin showed the largest inhibition zone (33 mm) against *Lb. plantarum* NCDO 995 (Figure 1b). Polylysine showed the largest inhibition zone (17 mm) against *M. luteus* ATCC 10420 (Figure 1c).

*E. coli* ATCC 12435, the non-pathogenic indicator organism used as the negative control, was not sensitive to any artisanal kefirs (Figure 1a). *M. luteus* and *Lb. plantarum*, non-pathogenic indicators, were sensitive to nisin, polylysine, and all artisanal kefirs (A, B, C, I, K, and L) used at a range of volumes (100–250 µL) (Figure 1b,c). *Lb. plantarum* inhibition zones observed with artisanal kefirs (250 µL) were 8.5–16 mm (Figure 1b). *M. luteus* inhibition zones observed with artisanal kefirs (250 µL) were 14–20.5 mm (Figure 1c). Results obtained with non-pathogenic indicators confirmed that experiments employing the agar well diffusion method worked well.

All artisanal kefirs, A, B, C, I, K, and L, showed significant (*p* < 0.002) antimicrobial activity against pathogenic indicators (Figure 1d–g). Kefirs A, B, and I inhibited all pathogenic indicators at different levels (*p* < 0.001) (Figure 1d–g). Kefirs C, K, and L inhibited all pathogenic indicators (Figure 1d–g), except *S. aureus* ATCC 25923 (Figure 1f), which was inhibited by kefirs A, B, and I, compared to the negative control *E. coli* ATCC 12435 (*p* < 0.05). *S. aureus* ATCC 25923 displayed inhibition zones with increasing kefir volumes (150–250 µL) with kefirs A, B, and I (Figure 1f). *S. enterica* serovar Enteritis ATCC 13076 was inhibited by all kefir types (*p* < 0.05), with inhibition zones of 7–10 mm at 250 µL volume. *L. monocytogenes* CWD 1198 was inhibited the most by kefir I, K, and B (*p* < 0.05), with an inhibition zone of 12.5–14 mm, and at a volume of 250 µL (Figure 1e). The antilisterial effect of kefir I and kefir K (used at 250 µL) was equal to the antilisterial activity of nisin (100 µL = 220 IU) (Figure 1e). *L. monocytogenes* CWD 1198 displayed increasing inhibition zones with increasing kefir volumes (150–250 µL) with kefirs A, B, C, I, and K but not kefir L (Figure 1e). With kefir L, *L. monocytogenes* CWD 1198 inhibition zone was constant (7 mm) regardless of the kefir L volume applied (Figure 1e). *B. cereus* showed the same inhibition zone (7 mm) with all artisanal kefirs, regardless of the kefir volumes used (100–250 µL), and with both positive bacteriocin controls (100 µL each) (Figure 1d).

### 3.3. Ruling-Out Any Antimicrobial Activity Due to Organic Acids, Hydrogen Peroxide, and Free Fatty Acids Produced in Artisanal Kefir

LAB are known to produce antibacterial metabolites, including bacteriocins, organic acids, H_2_O_2_, and fatty acids. Application of filter-sterilized artisanal kefir samples treated with proteinase K from *Tritirachium album*, β-glycerophosphate, bovine liver catalase, and lipase from *Aspergillus oryzae* in agar well diffusion experiments allowed us to rule out any antimicrobial activity due to these metabolites produced by LAB in kefir.

As anticipated, *E. coli* ATCC 12435, a non-pathogenic organism used as a negative control, was not sensitive to any artisanal kefir (A, I, and K), untreated or treated with proteinase K, β-glycerophosphate, catalase, and lipase. (Figure 2a–c). This confirms the antimicrobial activity results obtained for kefirs A, I, and K described in Section 3.2.2.

No bacterial inhibition zones were observed with *M. luteus* or any pathogenic indicators when proteinase K-treated kefirs A, I, and K were used (*p* < 0.001), confirming that the majority of the antibacterial activity observed was due to bacteriocins with proteinaceous nature (Figure 2a–c). As expected, proteinase K-treated nisin and polylysin, used as bacteriocin controls, did not show any inhibitory activity against *M. luteus* or any pathogenic indicators.

*M. luteus*, a non-pathogenic organism used as a positive control, was sensitive to all untreated kefirs (A, I, and K) and all catalase-, lipase-, and β-glycerophosphate-treated kefirs (A, I, and K) (Figure 2a–c). *M. luteus* inhibition zones observed with untreated kefir A were 9.5 mm while the catalase-, lipase-, and β-glycerophosphate-treated kefir A samples were 8, 8, and 11 mm, respectively, indicating that both H_2_O_2_ and free fatty acids made contributions to total antimicrobial activity (Figure 2a). The increase in antimicrobial activity with β-glycerophosphate-treated kefir A can be explained by an increase in bacteriocin(s) activity at the higher pH achieved with β-glycerophosphate addition to kefir A. While untreated kefir I resulted in an inhibition zone of 10.5 mm for *M. luteus*, the catalase-, lipase-, and β-glycerophosphate-treated kefir I samples resulted in an inhibition zone of 9.5 mm, which was not a significant difference (*p* > 0.05) (Figure 2b). *M. luteus* was inhibited by untreated kefir K, with an inhibition zone of 10 mm, as well as the catalase-, lipase-, and β-glycerophosphate-treated kefir K samples with inhibition zones of 8.5, 8, and 8 mm, respectively. These results show a very similar contribution to the total antimicrobial activity by H_2_O_2_, free fatty acids, and organic acids (Figure 2c).

The untreated and the catalase-, lipase-, and β-glycerophosphate-treated kefir A samples resulted in the inhibition zones of 8 and 7 mm for *S. aureus* and *B. cereus*, respectively, indicating that the bacteriocin activity is responsible for the antimicrobial activity against the two pathogenic indicator organisms (Figure 2a). While the untreated kefir A sample resulted in an inhibition zone of 10.5 mm against *L. monocytogenes*, the catalase-, lipase-, and β-glycerophosphate-treated kefirs resulted in the inhibition zones of 8, 8, and 11.5 mm, respectively (Figure 2a). These observations indicate that *L. monocytogenes* was inhibited mostly by bacteriocins in kefir A, but organic acids, free fatty acids, and H_2_O_2_ made contributions to the total inhibition. The individual contributions of free fatty acids and H_2_O_2_ to the total antimicrobial activity in kefir A were identical as per these results.

The bacteriocin activity in Kefir I was solely responsible for the antimicrobial activity observed against *B. cereus* (Figure 2b). Because, the untreated and the catalase-, lipase-, and β-glycerophosphate-treated kefir I samples resulted in the identical inhibition zones of 7 mm against *B. cereus.* In the case of *S. aureus*, the catalase-, lipase-, and β-glycerophosphate-treated kefir I samples resulted in the inhibition zone of 8 mm, which is 1 mm less than the inhibition zone (9mm) observed with untreated kefir I (Figure 2b). With *L. monocytogenes*, the catalase-, lipase-, and β-glycerophosphate-treated kefir I samples resulted in the inhibition zones of 7.5–8 mm, which are smaller than the inhibition zone (10 mm) observed with untreated kefir I (Figure 2b). These results indicated that organic acids, free fatty acids, and H_2_O_2_ made identical contributions to the total inhibition of the two pathogens. An identical inhibition zone (8 mm) was observed with *S. enterica* serovar Enteritidis when using the untreated and β-glycerophosphate-treated kefir I, indicating that the total antimicrobial activity is due to bacteriocins in kefir I (Figure 2b). The *S. enterica* serovar Enteritidis inhibition zones of 8.25 and 8.75 mm, observed with the lipase- and catalase-treated kefir I respectively, were slightly larger than the 8 mm zone observed with untreated kefir (Figure 2b). The difference was not statistically significant. 

The untreated and the catalase-, lipase-, and β-glycerophosphate-treated kefir K samples resulted in the inhibition zones of 8 and 7 mm for *S. enterica* serovar Enteritidis and *B. cereus,* respectively, illustrating that the bacteriocin activity in kefir K is solely responsible for the total antimicrobial activity against these two pathogens (Figure 2c). For *L. monocytogenes*, an inhibition zone of 8 mm was observed with the catalase-, lipase-, and β-glycerophosphate-treated kefir K samples while the untreated kefir K sample resulted in an inhibition zone of 9 mm (Figure 2c). Based on the results, the individual contribution of organic acids, free fatty acids, and H_2_O_2_ to total antimicrobial activity is identical. The untreated and the catalase-, lipase-, and β-glycerophosphate-treated kefir K samples did not exhibit any anti-staphylococcal activity (Figure 2c), confirming the results described for kefir K in Section 3.2.2.

## 4. Discussion

In the presented work, we explored the antimicrobial activity of international artisanal kefirs from Fusion Tea, Britain, the Caucuses region, Ireland, Lithuania, and South Korea and made comparisons among these kefirs regarding their antimicrobial activity. Based on our findings, antimicrobials with proteinaceous nature (e.g., bacteriocins) are responsible for the majority of antibacterial activity observed against the foodborne pathogens tested. To our knowledge, this is the first report in the literature that deals with such comparisons among international artisanal kefirs.

Our work targeted select foodborne pathogens, including *L. monocytogenes*, *S. enterica* serovar Enteritidis, *S. aureus*, and *B. cereus,* all known to be hazardous to human health. One of these pathogens, *L. monocytogenes,* is considered to be a major challenge for the food industry worldwide. In particular, refrigerated, ready-to-eat (RTE) foods pose a high listeriosis risk because refrigeration offers an environment in which *L. monocytogenes* can outcompete other mesophilic microorganisms. Another target pathogen, *B. cereus*, especially psychrotrophic strains, are problematic in dairy foods for various reasons: (1) *B. cereus* readily spread from healthy and decaying plants and soil to the cows and raw milk; (2) hydrophobic *B. cereus* spores attach to surfaces in dairy plants; *B. cereus* spores survive milk pasteurization, germinate in the absence of competitive microflora, and cause problems in milk products. Therefore, additional food preservation approaches need to be in place to ensure the safety of refrigerated foods. Biologically based preservation, through the use of LAB, can be used to enhance microbial food safety of refrigerated foods without modifying them. LAB are accepted by many countries in the world as “GRAS”. LAB are also perceived by consumers in the world as “natural,” “healthy,” and “health-promoting”.

LAB have been shown to produce bacteriocins that inhibit foodborne pathogens such as *L. monocytogenes* [22]. Kefir contains many LAB species known for their bacteriocin production and probiotic benefits [16,23,24]. Joao et al. tested kefir products from Brazil against *L. monocytogenes* ATCC 15313 and found 54.18% inhibition when compared to that of the untreated control [17]. Our research findings are in agreement with that of Joao et al. in that kefir has antilisterial activity. In our work, all artisanal kefirs (A, B, C, I, K, and L) exhibited varying levels of inhibition against *L. monocytogenes*.

Artisanal kefir has been shown to exhibit various antimicrobial activities against foodborne pathogens and spoilage organisms [17,18,19]. Coconut milk inoculated with kefir grains from India showed antimicrobial and antifungal activity against *E. coli*, *S. typhi*, *S. aureus*, *Saccharomyces cerevisiae,* and *Aspergillus niger* [25]. Kefir originating from Argentina inhibited the growth of *E. coli* ATCC 11229, *S. enterica* serovar Enteritidis CIDCA 101, and *B. cereus* ATCC 10876 [26]. In our study, artisanal kefirs originating from six different regions of the world showed antimicrobial activity against select foodborne pathogens. Kim et al. compared the antimicrobial spectra of four types of kefirs from South Korea, which showed inhibition against select strains of *B. cereus*, *S. enterica* serovar Enteritidis, *P. aeruginosa*, *C. sakazakii,* and *L. monocytogenes* [19]. We showed that the same South Korean kefir (K) inhibited all pathogenic indicator strains tested except *S. aureus* (*p* < 0.05). Our study supports the findings of Kim et al. in that South Korean kefir has a wide antimicrobial activity spectrum [19]. When compared to other kefirs in this study, the kefir from South Korea (K) showed the highest antimicrobial activity against *L. monocytogenes* when using the smallest volume (100 ul) of kefir product. The kefirs from Ireland, Britain, and Fusion Tea inhibited *S. aureus* (*p* < 0.05), in contrast to the South Korean kefir.

In our study, kefir K exhibited antimicrobial activity against all foodborne pathogenic indicators, except *S. aureus*. Consequently, our goal is to carry out additional research on kefir K with emphasis on isolation and characterization of LAB and their bacteriocins and application of these bacteriocins as natural antimicrobials against *L. monocytogenes, B. cereus,* and *S. enterica*. In addition, kefir A and I showed the highest inhibition zones against *S. aureus*. Therefore, we are interested in further studying kefir A and kefir I as sources of natural antimicrobials against *S. aureus.*

## 5. Conclusions

Consumers demand natural, health-promoting, and nutritious food. Chemical additives have been extensively used in food preservation but their safety and impact on human health continue to be under discussion. The food industry desires to replace chemical preservatives with natural biopreservatives. Kefir has a natural antimicrobial activity due to the presence of LAB with bacteriocin production capability. In this study of artisanal kefirs from different countries, we have elucidated that bacteriocin production is the main reason for these kefirs’ antimicrobial activity against select foodborne pathogens. Based on our findings in this study, kefir-based antimicrobials are being explored in our laboratory as promising natural biopreservatives in model food systems. 

## Figures and Tables

**Figure 1 microorganisms-08-00849-f001:**
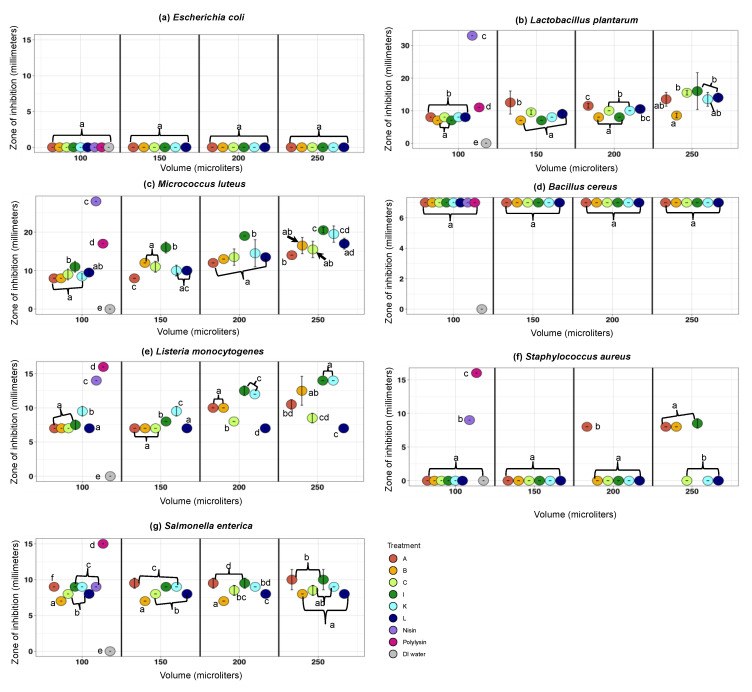
Comparison of the antimicrobial activity spectra of artisanal kefirs from Fusion Tea (A, Amazon); Britain (B); the Caucasus region (C); Ireland (I); South Korea (K); and Lithuania (L) against (**a**) *Escherichia coli* ATCC 12435, (**b**) *Lactobacillus plantarum* NCDO 995, (**c**) *Micrococcus luteus* ATCC 10420, (**d**) *Bacillus cereus* ATCC 14579, (**e**) *Listeria monocytogenes* CWD 1198, (**f**) *Staphylococcus aureus* ATCC 25923, and (**g**) *Salmonella enterica* serovar Enteritidis ATCC 13076 using the agar well diffusion method. A range of filter-sterilized artisanal kefir volumes (100 µL, 150 µL, 200 µL, and 250 µL) were placed inside the wells. Nisin (N; 220 IU) and polylysin (P; 200 IU) were used as positive controls in a volume of 100 µL. Sterilized DI water (100 µL) was used as a negative control. The diameter of the inhibition zones was measured in mm. All experiments were conducted two independent times and each time in duplicate. Different letters (**a**–**f**) indicate statistical pairwise comparisons between the treatments within each volume performed by post-hoc Tukey’s multiple comparison procedure. The same letter indicates no significant difference between the treatments within each volume.

**Figure 2 microorganisms-08-00849-f002:**
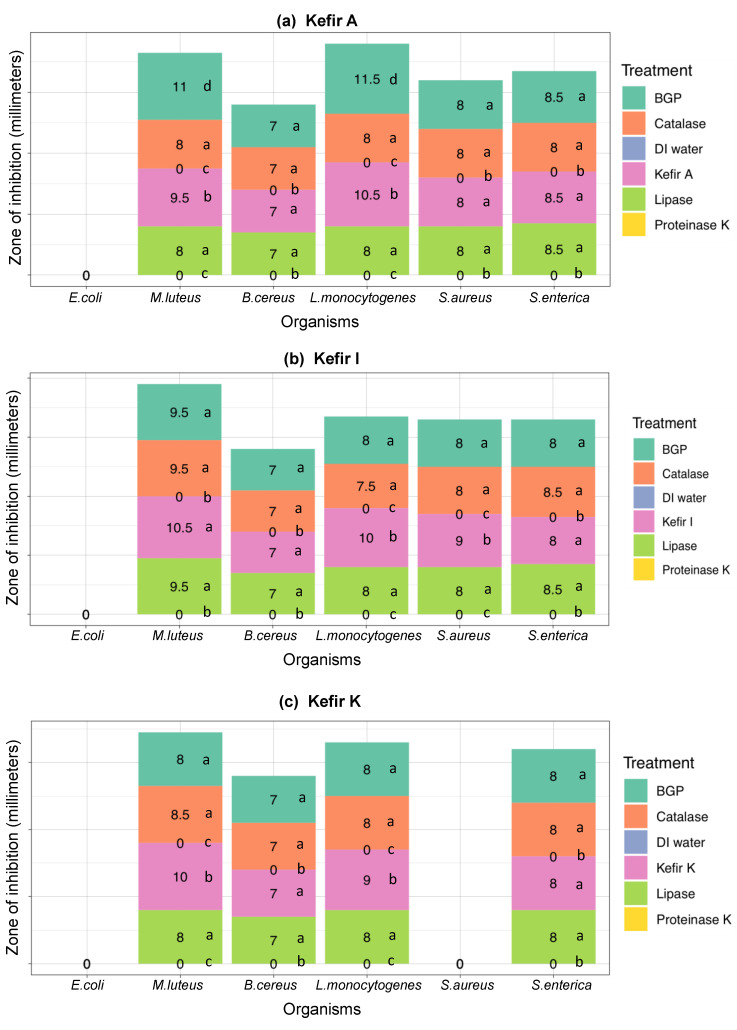
Bacteriocin-based antimicrobial activity in filter-sterilized artisanal kefir samples from (**a**) Fusion Tea (Kefir A), (**b**) Ireland (Kefir I), and (**c**) South Korea (Kefir K) against select indicator strains using β-glycerophosphate (BGP, teal), bovine liver catalase (coral), lipase from *Aspergillus oryzae* (lime green), and proteinase K from *Tritirachium album* (yellow). The agar well diffusion method was used. Sterile DI water (blue) was used as the negative control. Wells generated in each plate contained filter-sterilized artisanal kefir product (100 µL) with or without the additives mentioned above. The diameter of the inhibition zones was measured in mm after incubation for 24 h at 37 or 30 °C. Different letters (**a**–**c**) indicate statistical pairwise comparisons between the treatments for a given organism performed by post-hoc Tukey’s multiple comparison procedure. The same letter indicates no significant difference between treatments for a given organism.

**Table 1 microorganisms-08-00849-t001:** Artisanal kefir origin, source, grains’ description and products’ description.

Kefir Origin	Source	Grains’ Description	Products’ Description
Lithuania	Etsy Inc.	Cauliflower-like appearance, off-white to pale yellow, medium size (1–10 mm) and firm grains	Mild, smooth, and not sour (sweet)
Ireland	Etsy Inc.	Soft, small size (>1 mm) grains	Mild, sweet and pleasant taste, smooth, sweet aroma, fresh, and cheesy
The Caucuses region	Etsy Inc.	Cauliflower-like appearance, off-white to pale yellow, size 2–10 mm, firm, rubbery with smooth grains	Earthy, cheesy aroma, and sour taste
South Korea	[19]	Soft, curling, size 2–10 mm	Earthy, cheesy aroma, and sour taste
Britain	Etsy Inc.	Cauliflower-like appearance, small to large size (2.5–50 mm), rubbery, firm, smooth grains	Creamy, earthy, cheesy aroma, slightly sour
Fusion Tea	Amazon	Cauliflower-like appearance, off-white to pale yellow, mixed sizes (2–7 mm), firm, rubbery textured grains	Smooth, mild sour, creamy, pleasant, and fresh, sweet, yeasty aroma

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
