# Peer review of "Antimicrobial Activity of Six International Artisanal Kefirs against Bacillus cereus, Listeria monocytogenes, Salmonella enterica Serovar Enteritidis, and Staphylococcus aureus"

_microorganisms, 2020, doi:10.3390/microorganisms8060849_

Round 1
Reviewer 1 Report
The paper contains a lot of nice results, actually I would split it in two, because the second part, referring to the microbial diversity of kefirs is not well integrated with the first one.
The authors evaluated the microbial diversity of the tested kefirs, but did not comment how the abundance of a certain species is related with a certain spectrum of antimicrobial activity.
In the first part of the manuscript, the graphic presentation of the obtained results could be improved, the figures presenting the antimicrobial activity are very difficult to follow. The present figures could be moved to supplemental and a more synthetic, attractive way of results presentation needs to be found. For example, for each bacterial species, the efficiency could be presented in descending order.
The abbreviation S. aureus needs to be used all over the manuscript, instead of Staph. aurerus.
Author Response
Dear Reviewer,
Thank you for your careful review of our article entitled “Antimicrobial activity of six international artisanal kefirs against Bacillus cereus, Listeria monocytogenes, Salmonella enterica serovar Enteritidis, and Staphylococcus aureus and their microbial diversity.” We appreciate your helpful comments and suggestions. We believe that the second version of our manuscript is an improved version.
Comments and Suggestions for Authors
The paper contains a lot of nice results, actually I would split it in two, because the second part, referring to the microbial diversity of kefirs is not well integrated with the first one.
The authors evaluated the microbial diversity of the tested kefirs, but did not comment how the abundance of a certain species is related with a certain spectrum of antimicrobial activity.
Thank you for your comment on our “paper containing a lot of nice results” and your suggestion to split the paper. We tried to link the antimicrobial activity observed in artisanal kefir products to lactic acid bacterial species with known bacteriocin production capabilities under section “3.4 Genetic approaches for identification of bacteria in kefir samples”. Your comments indicate that we were not completely successful in our attempt. Therefore, we have split the paper into two as per your invaluable suggestion. We plan to draft a second manuscript on exploring microbial diversity of international artisanal kefirs through 16S rRNA sequencing for likely submission to the Journal of Microorganisms.
In the first part of the manuscript, the graphic presentation of the obtained results could be improved, the figures presenting the antimicrobial activity are very difficult to follow. The present figures could be moved to supplemental and a more synthetic, attractive way of results presentation needs to be found. For example, for each bacterial species, the efficiency could be presented in descending order.
Thank you for pointing this out to us. Using ggplot2, we made drastic changes to Figure 1 and 2. We believe that the figures are now modern, attractive and easy-to-follow. We hope you will like the new figures.
The abbreviation S. aureus needs to be used all over the manuscript, instead of Staph. aurerus.
We used Staph. aureus to differentiate the genera Salmonella (S.), Streptococcus (Strep.), and Staphylococcus (Staph.). We changed Staph. aureus to S. aureus throughout the manuscript upon your request. Thank you.

Reviewer 2 Report
The paper “Antimicrobial activity of six international artisanal 2 kefirs against Bacillus cereus, Listeria monocytogenes, 3 Salmonella enterica serovar Enteritidis, and 4 Staphylococcus aureus and their microbial diversity” focuses on the microbial diversity of kefir grains from various locations and examines their antimicrobial activity towards some pathogens.
Although the topic could be of interest, the paper does not deserve publication, at least in its present form.
There are some major issues, mainly linked to methodology and statistic, strongly limiting the impact of the paper. The authors investigate the antimicrobial activity of kefir and their hypothesis is that the effect could be the result of some compounds (organic acids, bacteriocins etc…); however, kefir is not sterile and the activity could be also linked to the interactions occurring between kefir microbiota and target pathogens or to the combination microbiota/antimicrobial compounds. With the methodology of this paper, the authors cannot exclude it:
The correct approach could be as follows:
- Control (only pathogens)
- Control with a known antibiotic (pathogens+antibiotic)
- Kefir (pathogens+kefir)
- Inactivated kefir (kefir inactivated to kill microorganisms+pathogens)
- Controls with the various compounds, as reported in the paper (acid, bacteriocin etc…+pathogens)
This approach would result in a complete scheme on the possible mode of action of kefir, as well as to point out the amount of effect due to kefir microbiota and that due to the compounds.
Another issue is on the amounts of compounds used: how were the concentrations of acids, enzymes chosen? Did they reflect the effective amounts in kefir?
Finally, statistic is a drawback; there is not statistic on the first part of the research, therefore it not possible to assess the significance of the results, as well as differences amongst pathogens or grains.
Author Response
Dear Reviewer,
Thank you for your careful review of our article entitled “Antimicrobial activity of six international artisanal kefirs against Bacillus cereus, Listeria monocytogenes, Salmonella enterica serovar Enteritidis, and Staphylococcus aureus and their microbial diversity.” We appreciate your helpful comments and suggestions. We believe that the second version of our manuscript is an improved version.
Please note that we have split the paper into two as per Reviewer 1's recommendation. We plan to draft a second manuscript on exploring microbial diversity of international artisanal kefirs through 16S rRNA sequencing for likely submission to the Journal of Microorganisms.
Comments and Suggestions for Authors
The paper “Antimicrobial activity of six international artisanal 2 kefirs against Bacillus cereus, Listeria monocytogenes, 3 Salmonella enterica serovar Enteritidis, and 4 Staphylococcus aureus and their microbial diversity” focuses on the microbial diversity of kefir grains from various locations and examines their antimicrobial activity towards some pathogens.
Although the topic could be of interest, the paper does not deserve publication, at least in its present form.
There are some major issues, mainly linked to methodology and statistic, strongly limiting the impact of the paper. The authors investigate the antimicrobial activity of kefir and their hypothesis is that the effect could be the result of some compounds (organic acids, bacteriocins etc…); however, kefir is not sterile and the activity could be also linked to the interactions occurring between kefir microbiota and target pathogens or to the combination microbiota/antimicrobial compounds. With the methodology of this paper, the authors cannot exclude it:
The correct approach could be as follows:
- Control (only pathogens)
- Control with a known antibiotic (pathogens+antibiotic)
- Kefir (pathogens+kefir)
- Inactivated kefir (kefir inactivated to kill microorganisms+pathogens)
- Controls with the various compounds, as reported in the paper (acid, bacteriocin etc…+pathogens)
This approach would result in a complete scheme on the possible mode of action of kefir, as well as to point out the amount of effect due to kefir microbiota and that due to the compounds
Thank you for your comments and suggestions. We believe there is some miscommunication that requires clarification. We always worked with filter-sterilized kefir samples. The following statements, which illustrate the use of filter-sterilized kefir samples, were included in our original manuscript. The first statement (line 145-147) under section 2.1 mentions the use of a Millex filter to sterilize kefir (please see below). The second statement (line 186) refers to section 2.1., which mentions the use of Millex filter (please see below). The third statement (lines 214-217) mentions the use of sterile kefir samples (please see below). The fourth statement (line 321-324) mentions the application of filter-sterilized artisanal kefir samples treated with proteinase K, β-glycerophosphate, bovine liver catalase, and lipase from Aspergillus oryzae in agar well diffusion experiments. We hope our explanation offers sufficient clarification in this regard. We have included the term “filter-sterilized” in a few key areas for additional clarity.
Statement 1:
2.1. Kefir preparation for determining kefir antimicrobial activity
…Filtration through a Millex filter (Millipore Corporation, Bedford, MA, USA) with 45 µm pore size was used to sterilize the samples. The kefir samples were tested for their antimicrobial activities immediately.”
Statement 2:
2.4. Detection of antimicrobial activity
“…Kefir samples were obtained as described above.”
Statement 3:
2.5.3. Detection of antimicrobial activity due to bacteriocin production in kefir
“…Sterile bovine liver catalase (2000-5000 U/mg protein), Aspergillus oryzae lipase (≥ 100,000 U/g), Proteinase K (39 U/mg), and β-glycerophoshate were added to sterile kefir samples A, K and I at a final concentration of 1 mg/ml….”
Statement 4:
3.3 Ruling out any antimicrobial activity due to organic acids, hydrogen peroxide and free fatty acids produced in kefir
“LAB are known to produce antibacterial metabolites, including bacteriocins, organic acids, H2O2, and fatty acids. Application of filter-sterilized artisanal kefir samples treated with proteinase K, β-glycerophoshate, bovine liver catalase, and lipase from Aspergillus oryzae in agar well diffusion experiments allowed us to rule out any antimicrobial activity due to these metabolites produced by LAB in kefir.”
Another issue is on the amounts of compounds used: how were the concentrations of acids, enzymes chosen? Did they reflect the effective amounts in kefir?
Thank you for your comment.
In our work, sterile bovine liver catalase (2000-5000 U/mg protein), Aspergillus oryzae lipase (≥ 100,000 U/g), and Proteinase K from Tritirachium album (≥30 units/mg) were added to filter-sterilized pH-adjusted (6.0) kefir samples A, K and I at a final concentration of 1 mg/ml. β-glycerophosphate was added to filter-sterilized pH-adjusted (6.0) kefir samples A, K and I at 1% (w/v). These mixtures were incubated at 37oC for 1 h prior to being applied in the agar well diffusion experiments. The final concentrations of 1% and 1 mg/ml have been used in several papers that deal with ruling out any antimicrobial activity due to organic acids and other metabolites (hydrogen peroxide and free fatty acids), respectively, produced by lactic acid bacteria. In our case, we followed and cited Dimitrieva-Moats and Ünlü (2011), Ünlü et al. (2015), and Ünlü et al. (2016).
We used Proteinase K from Tritirachium album (lyophilized powder, BioUltra, ≥30 units/mg protein, for molecular biology) at a final concentration of 1mg/ml (= ≥30 units/ml). Other research on bacteriocins have stated the use of Proteinase K at the same (1 mg/ml) concentration (Powell, 2006). In our work, no bacterial inhibition zones were observed with Micrococcus luteus or any pathogenic indicators when proteinase K-treated artisanal kefirs A, I, and K were used. Bacterial inhibition zones were observed with untreated artisanal kefirs A, I, and K with Micrococcus luteus and pathogenic indicators. These results indicate that the proteinase K treatment we applied was successful in degrading bacteriocins with proteinaceous nature.
We used catalase from bovine liver (lyophilized powder, 2,000-5,000 units/mg protein) at a final concentration of 1 mg/ml (2,000-5,000 units/ml). According to the manufacturer (Sigma Aldrich) of bovine liver catalase, the unit definition for the enzyme is: “One unit will decompose 1.0 μmole of H2O2 per min at pH 7.0 at 25 °C, while the H2O2 concentration falls from 10.3 to 9.2 mM, measured by the rate of decrease of A240.”
There is limited literature on H2O2 concentration in kefir. Kesenkas et al. (2011) determined antioxidant properties of kefir samples produced from different cow/soymilk mixtures, including hydrogen peroxide levels of kefir samples. They reported that none of their kefir samples exhibited hydrogen peroxide scavenging activity. Kivanc and Yapici (2015) isolated lactic acid bacteria from kefir samples. Lactobacillus paracasei spp paracasei KM-5 produced the maximum hydrogen peroxide (0.69 μg/mL).
In our work, sterile bovine liver catalase (2000-5000 U/mg protein) was added to filter-sterilized pH-adjusted (6.0) artisanal kefir samples A, K and I at a final concentration of 1 mg/ml. We have not measured hydrogen peroxide levels in our artisanal kefirs. However, the concentration (1 mg/ml = 2000-5000 U/ml) of catalase we used is capable of decomposing 2000-5000 micromol hydrogen peroxide per min. Hydrogen peroxide has a molecular weight of ~34 g/mol (0.034 microgram/micromol). In a hypothetical kefir sample that contains 20-50 micromol (0.69-1.7 microgram) hydrogen peroxide, 2000-5000 micromol catalase offers 100X decomposition power.
We used Aspergillus oryzae lipase (≥ 100,000 U/g) at a final concentration of 1 mg/ml. Other research on bacteriocins have stated the use of lipase at the same (1 mg/ml) concentration (Sharma et al., 2020). In addition, our findings showed higher (and statistically significant) antimicrobial activity in filter-sterilized pH-adjusted (6.0) artisanal kefir samples when compared to their filter-sterilized, pH-adjusted (6.0), and lipase-treated counterparts. This indicates that free fatty acids make a contribution to total antimicrobial activity in artisanal kefirs. This also indicates that the lipase treatment we applied was successful in degrading fatty acids in artisanal kefirs.
β-glycerophosphate has been used in previous research at a final concentration of 1 % (Ünlü et al. 025) to 2% (Zhan 2017) to neutralize organic acids produced by lactic acid bacteria. We used β-glycerophosphate at 1% to buffer filter-sterilized pH-adjusted (6.0) artisanal kefir samples. Our β-glycerophosphate treatment worked because β-glycerophosphate-treated artisanal kefir samples were at a pH range of 6.3-6.5 as determined by pH measurements over a 24 h incubation period at 30oC and 37oC. In addition, our findings showed higher (and statistically significant) antimicrobial activity in filter-sterilized pH-adjusted (6.0) artisanal kefir samples when compared to their filter-sterilized, pH-adjusted (6.0), and β-glycerophosphate-treated counterparts. This indicates that organic acids make a contribution to total antimicrobial activity in artisanal kefirs. This also indicates that the β-glycerophosphate treatment we applied was successful in buffering/neutralizing organic acids in artisanal kefirs.
Finally, statistic is a drawback; there is not statistic on the first part of the research, therefore it not possible to assess the significance of the results, as well as differences amongst pathogens or grains.
It appears that there is miscommunication regarding the statistical analyses performed. The following section (lines 258-264) was included in the manuscript. We hope this section clarifies your concerns.
“2.7 Statistical analysis
For the detection of the antimicrobial activity of kefir, the experiment was a completely random design. Three-way ANOVA was used for all of the following: indicator organisms, kefir types, kefir volumes and their interactions, followed by Tukey’s test for multiple comparison procedure using R (R Studio Inc., Boston, MA, USA) (p < 0.05). For ruling out any antimicrobial activity due to organic acids, hydrogen peroxide and free fatty acids produced in kefir, three-way ANOVA was used as well, followed by Tukey’s test for multiple comparison procedure using R (p < 0.05).”
References
Kesenkas, H.; Dinkci, N.; Seckin, K.; Kinik, O.; & Gonc, S. Antioxidant Properties of Kefir Produced from Different Cow and Soy Milk Mixtures. Journal of Agricultural Sciences 2011, 17, 253‐259. DOI:10.1501/tbd.v17i3.659
Kivanc, M.; Yapıcı, E. Kefir as a Probiotic Dairy Beverage: Determination Lactic Acid Bacteria and Yeast. International Journal of Food Engineering 2015, 1 (1), 55-60. DOI: 10.18178/ijfe.1.1.55-60
Powell, J. Bacteriocins and bacteriocin producers present in kefir and kefir grains. Thesis, M.Sc. Food Science, University of Stellenbosch 2006
Sharma, G.; Gupta, H.; Dang, S.; Gupta, S.; Gabrani, R. Characterization of antimicrobial substance with antibiofilm ability from Pediococcus acidilactici. The Journal of Microbiology, Biotechnology and Food Sciences 2020, 9 (5), 979-982. DOI: 10.15414/jmbfs.2020.9.5.979-982
Ünlü, G.; Nielsen, B.; Ionita, C., Production of Antilisterial Bacteriocins from Lactic Acid Bacteria in Dairy-Based Media: A Comparative Study. Probiotics Antimicrob Proteins 2015, 7 (4), 259-74. DOI: 10.1007/s12602-015-9200-z
Dimitrieva-Moats, G. Y.; Ünlü, G., Development of Freeze-Dried Bacteriocin-Containing Preparations from Lactic Acid Bacteria to Inhibit Listeria monocytogenes and Staphylococcus aureus. Probiotics Antimicrob Proteins 2012, 4 (1), 27-38. DOI: 10.1007/s12602-011-9088-1
Ünlü, G.; Nielsen, B.; Ionita, C., Inhibition of Listeria monocytogenes in Hot Dogs by Surface Application of Freeze-Dried Bacteriocin-Containing Powders from Lactic Acid Bacteria. Probiotics Antimicrob Proteins 2016, 8 (2), 102-10. DOI: 10.1007/s12602-016-9213-2
Zhang, J.; Yang, Y.; Yang, H.; Bu, Y.; Yi, H.; Zhang, L.; Han, X.; Ai, L. Purification and Partial Characterization of Bacteriocin Lac-B23, a Novel Bacteriocin Production by Lactobacillus plantarum J23, Isolated From Chinese Traditional Fermented Milk. Frontiers in Microbiology 2018, 9, 2165. https://doi.org/10.3389/fmicb.2018.02165

Round 2
Reviewer 2 Report
I have again a concern on statistic; I saw the section in the first version, but probably my issue was not clear. Where are the results of statistic in the figure? I would like to see letters to indicate significant differences.
Author Response
"I have again a concern on statistic; I saw the section in the first
version, but probably my issue was not clear. Where are the results of
statistic in the figure? I would like to see letters to indicate
significant differences.
What I mean is that there are two details to add in the figures:
a) Standard deviation
b) Letters of ANOVA.
The authors wrote that they performed statistic; if it is true, in
figures (for example in Fig. 1) add letters to highlight significant
differences for each microorganism and among microorganisms. Each
software, after performing a post-hoc test (Tukey, Fisher etc...) there
is a table showing the differences among samples. These differences are
generally shown as letters (samples with the same letter are not
significantly different).
In Figure 2, I can see frequencies. Are the differences significant?
Please perform Chi-square."
Dear Reviewer,
Thank you for your careful review of our article entitled “Antimicrobial activity of six international artisanal kefirs against Bacillus cereus, Listeria monocytogenes, Salmonella enterica serovar Enteritidis, and Staphylococcus aureus.” Our response to your helpful comments and suggestions are as follow:
Figure 1:
We added (a) Standard deviation, and (b) Letters of ANOVA (performed a post-hoc Tukey test) for Figure 1 in the revised manuscript. The statistically significant difference was determined by cut-off for significance level at 5% (i.e., p < 0.05). Please see the newly designed Figure 1 in our revised manuscript.
Also, we included the following statement within the figure legend for Figure 1: Different letters (a-f) indicate statistical pairwise comparisons between the treatments within each volume performed by post-hoc Tukey’s multiple comparison procedure (the same letter indicates no significant difference between the treatments within each volume).
Figure 2:
To test the statistically significant differences between the treatments within each organism, we performed post-hoc Tukey’s multiple comparison test and added letters of ANOVA in Figure 2 in the revised manuscript. The statistically significant difference was determined by cut-off for significance level at 5% (i.e., p < 0.05). Please see the newly designed Figure 2 in our revised manuscript.
We are unable to Chi-square test. Because, the frequencies shown in Figure 2 are the mean values of the treatments within each organism, and much of the data value has only one level. Therefore, we performed Tukey test to determine statistically significant differences.
Also, we included the following statement within the figure legend for Figure 2: Different letters (a-c) indicate statistical pairwise comparisons between the treatments within each organism performed by post-hoc Tukey’s multiple comparison procedure (the same letter indicates no significant difference between the treatments within each organism).
